# Theoretical Comparisons of Positive-Unlabeled Learning against Positive-Negative Learning

**Gang Niu**[1]   **Marthinus C. du Plessis**[1]   **Tomoya Sakai**[1]   **Yao Ma**[3]   **Masashi Sugiyama**[2,1]
[1]The University of Tokyo, Japan   [2]RIKEN, Japan   [3]Boston University, USA
{ gang@ms., christo@ms., sakai@ms., yao@ms., sugi@ }k.u-tokyo.ac.jp

## Abstract

In PU learning, a binary classifier is trained from *positive* (P) and *unlabeled* (U) data without *negative* (N) data. Although N data is missing, it sometimes outperforms PN learning (i.e., ordinary supervised learning). Hitherto, neither theoretical nor experimental analysis has been given to explain this phenomenon. In this paper, we theoretically compare PU (and NU) learning against PN learning based on the upper bounds on *estimation errors*. We find simple conditions when PU and NU learning are likely to outperform PN learning, and we prove that, in terms of the upper bounds, either PU or NU learning (depending on the class-prior probability and the sizes of P and N data) given infinite U data will improve on PN learning. Our theoretical findings well agree with the experimental results on artificial and benchmark data even when the experimental setup does not match the theoretical assumptions exactly.

## 1   Introduction

*Positive-unlabeled* (PU) learning, where a binary classifier is trained from P and U data, has drawn considerable attention recently [1, 2, 3, 4, 5, 6, 7, 8]. It is appealing to not only the academia but also the industry, since for example the click-through data automatically collected in search engines are highly PU due to position biases [9, 10, 11]. Although PU learning uses no *negative* (N) data, it is sometimes even better than PN learning (i.e., ordinary supervised learning, perhaps with class-prior change [12]) in practice. Nevertheless, there is neither theoretical nor experimental analysis for this phenomenon, and it is still an open problem when PU learning is likely to outperform PN learning. We clarify this question in this paper.

**Problem settings**   For PU learning, there are two problem settings based on *one sample* (OS) and *two samples* (TS) of data respectively. More specifically, let $X \in \mathbb{R}^d$ and $Y \in \{\pm 1\}$ ($d \in \mathbb{N}$) be the input and output random variables and equipped with an *underlying joint density* $p(x, y)$. In OS [3], a set of U data is sampled from the *marginal density* $p(x)$. Then if a data point $x$ is P, this P label is observed with probability $c$, and $x$ remains U with probability $1 - c$; if $x$ is N, this N label is never observed, and $x$ remains U with probability 1. In TS [4], a set of P data is drawn from the *positive marginal density* $p(x \mid Y = +1)$ and a set of U data is drawn from $p(x)$. Denote by $n_+$ and $n_u$ the sizes of P and U data. As two random variables, they are fully independent in TS, and they satisfy $n_+/(n_+ + n_u) \approx c\pi$ in OS where $\pi = p(Y = +1)$ is the *class-prior probability*. Therefore, TS is slightly more general than OS, and we will focus on TS problem settings.

Similarly, consider TS problem settings of PN and NU learning, where a set of N data (of size $n_-$) is sampled from $p(x \mid Y = -1)$ independently of the P/U data. For PN learning, if we enforce that $n_+/(n_+ + n_-) \approx \pi$ when sampling the data, it will be ordinary supervised learning; otherwise, it is supervised learning with *class-prior change*, a.k.a. *prior probability shift* [12].

In [7], a *cost-sensitive formulation* for PU learning was proposed, and its risk estimator was proven *unbiased* if the surrogate loss is *non-convex* and satisfies a *symmetric condition*. Therefore, we can naturally compare empirical risk minimizers in PU and NU learning against that in PN learning.

**Contributions**   We establish risk bounds of three risk minimizers in PN, PU and NU learning for comparisons in a flavor of *statistical learning theory* [13, 14]. For each minimizer, we firstly derive a *uniform deviation bound* from the risk estimator to the risk using *Rademacher complexities* (see, e.g., [15, 16, 17, 18]), and secondly obtain an *estimation error bound*. Thirdly, if the surrogate loss is *classification-calibrated* [19], an *excess risk bound* is an immediate corollary. In [7], there was a *generalization error bound* similar to our uniform deviation bound for PU learning. However, it is based on a tricky decomposition of the risk, where surrogate losses for risk minimization and risk analysis are different and labels of U data are needed for risk evaluation, so that no further bound is implied. On the other hand, ours utilizes the same surrogate loss for risk minimization and analysis and requires no label of U data for risk evaluation, so that an estimation error bound is possible.

Our main results can be summarized as follows. Denote by $\hat{g}_{\mathrm{pn}}$, $\hat{g}_{\mathrm{pu}}$ and $\hat{g}_{\mathrm{nu}}$ the risk minimizers in PN, PU and NU learning. Under a mild assumption on the function class and data distributions,

- Finite-sample case: The estimation error bound of $\hat{g}_{\mathrm{pu}}$ is tighter than that of $\hat{g}_{\mathrm{pn}}$ whenever $\pi/\sqrt{n_+} + 1/\sqrt{n_{\mathrm{u}}} < (1-\pi)/\sqrt{n_-}$, and so is the bound of $\hat{g}_{\mathrm{nu}}$ tighter than that of $\hat{g}_{\mathrm{pn}}$ if $(1-\pi)/\sqrt{n_-} + 1/\sqrt{n_{\mathrm{u}}} < \pi/\sqrt{n_+}$.
- Asymptotic case: Either the *limit of bounds* of $\hat{g}_{\mathrm{pu}}$ or that of $\hat{g}_{\mathrm{nu}}$ (depending on $\pi$, $n_+$ and $n_-$) will improve on that of $\hat{g}_{\mathrm{pn}}$, if $n_+, n_- \to \infty$ in the same order and $n_{\mathrm{u}} \to \infty$ faster in order than $n_+$ and $n_-$.

Notice that both results rely on only the constant $\pi$ and variables $n_+$, $n_-$ and $n_{\mathrm{u}}$; they are simple and independent of the specific forms of the function class and/or the data distributions. The asymptotic case is from the finite-sample case that is based on theoretical comparisons of the aforementioned upper bounds on the estimation errors of $\hat{g}_{\mathrm{pn}}$, $\hat{g}_{\mathrm{pu}}$ and $\hat{g}_{\mathrm{nu}}$. To the best of our knowledge, this is the first work that compares PU learning against PN learning.

Throughout the paper, we assume that the class-prior probability $\pi$ is known. In practice, it can be effectively estimated from P, N and U data [20, 21, 22] or only P and U data [23, 24].

**Organization**   The rest of this paper is organized as follows. Unbiased estimators are reviewed in Section 2. Then in Section 3 we present our theoretical comparisons based on risk bounds. Finally experiments are discussed in Section 4.

## 2   Unbiased estimators to the risk

For convenience, denote by $p_+(x) = p(x \mid Y = +1)$ and $p_-(x) = p(x \mid Y = -1)$ partial marginal densities. Recall that instead of data sampled from $p(x, y)$, we consider three sets of data $\mathcal{X}_+$, $\mathcal{X}_-$ and $\mathcal{X}_{\mathrm{u}}$ which are drawn from three marginal densities $p_+(x)$, $p_-(x)$ and $p(x)$ independently.

Let $g : \mathbb{R}^d \to \mathbb{R}$ be a *real-valued decision function* for binary classification and $\ell : \mathbb{R} \times \{\pm 1\} \to \mathbb{R}$ be a *Lipschitz-continuous loss function*. Denote by

$$R_+(g) = \mathbb{E}_+[\ell(g(X), +1)], \quad R_-(g) = \mathbb{E}_-[\ell(g(X), -1)]$$

partial risks, where $\mathbb{E}_\pm[\cdot] = \mathbb{E}_{X \sim p_\pm}[\cdot]$. Then the *risk of g w.r.t. $\ell$ under $p(x, y)$* is given by

$$R(g) = \mathbb{E}_{(X,Y)}[\ell(g(X), Y)] = \pi R_+(g) + (1-\pi)R_-(g). \tag{1}$$

In PN learning, by approximating $R(g)$ based on Eq. (1), we can get an *empirical risk estimator* as

$$\widehat{R}_{\mathrm{pn}}(g) = \tfrac{\pi}{n_+} \sum_{x_i \in \mathcal{X}_+} \ell(g(x_i), +1) + \tfrac{1-\pi}{n_-} \sum_{x_j \in \mathcal{X}_-} \ell(g(x_j), -1).$$

For any fixed $g$, $\widehat{R}_{\mathrm{pn}}(g)$ is an *unbiased* and *consistent* estimator to $R(g)$ and its convergence rate is of order $\mathcal{O}_p(1/\sqrt{n_+} + 1/\sqrt{n_-})$ according to the *central limit theorem* [25], where $\mathcal{O}_p$ denotes the order in probability.

In PU learning, $\mathcal{X}_-$ is not available and then $R_-(g)$ cannot be directly estimated. However, [7] has shown that we can estimate $R(g)$ without any bias if $\ell$ satisfies the following *symmetric condition*:

$$\ell(t, +1) + \ell(t, -1) = 1. \tag{2}$$

Specifically, let $R_{\mathrm{u},-}(g) = \mathbb{E}_X[\ell(g(X), -1)] = \pi\mathbb{E}_+[\ell(g(X), -1)] + (1-\pi)R_-(g)$ be a risk that U data are regarded as N data. Given Eq. (2), we have $\mathbb{E}_+[\ell(g(X), -1)] = 1 - R_+(g)$, and hence

$$R(g) = 2\pi R_+(g) + R_{\mathrm{u},-}(g) - \pi. \tag{3}$$

By approximating $R(g)$ based on (3) using $\mathcal{X}_+$ and $\mathcal{X}_\mathrm{u}$, we can obtain

$$\widehat{R}_{\mathrm{pu}}(g) = -\pi + \tfrac{2\pi}{n_+}\sum_{x_i\in\mathcal{X}_+}\ell(g(x_i), +1) + \tfrac{1}{n_\mathrm{u}}\sum_{x_j\in\mathcal{X}_\mathrm{u}}\ell(g(x_j), -1).$$

Although $\widehat{R}_{\mathrm{pu}}(g)$ regards $\mathcal{X}_\mathrm{u}$ as N data and aims at separating $\mathcal{X}_+$ and $\mathcal{X}_\mathrm{u}$ if being minimized, it is an unbiased and consistent estimator to $R(g)$ with a convergence rate $\mathcal{O}_p(1/\sqrt{n_+} + 1/\sqrt{n_\mathrm{u}})$ [25].

Similarly, in NU learning $R_+(g)$ cannot be directly estimated. Let $R_{\mathrm{u},+}(g) = \mathbb{E}_X[\ell(g(X), +1)] = \pi R_+(g) + (1-\pi)\mathbb{E}_-[\ell(g(X), +1)]$. Given Eq. (2), $\mathbb{E}_-[\ell(g(X), +1)] = 1 - R_-(g)$, and

$$R(g) = R_{\mathrm{u},+}(g) + 2(1-\pi)R_-(g) - (1-\pi). \tag{4}$$

By approximating $R(g)$ based on (4) using $\mathcal{X}_\mathrm{u}$ and $\mathcal{X}_-$, we can obtain

$$\widehat{R}_{\mathrm{nu}}(g) = -(1-\pi) + \tfrac{1}{n_\mathrm{u}}\sum_{x_i\in\mathcal{X}_\mathrm{u}}\ell(g(x_i), +1) + \tfrac{2(1-\pi)}{n_-}\sum_{x_j\in\mathcal{X}_-}\ell(g(x_j), -1).$$

**On the loss function**   In order to train $g$ by minimizing these estimators, it remains to specify the loss $\ell$. The *zero-one loss* $\ell_{01}(t, y) = (1 - \mathrm{sign}(ty))/2$ satisfies (2) but is non-smooth. [7] proposed to use a *scaled ramp loss* as the surrogate loss for $\ell_{01}$ in PU learning:

$$\ell_{\mathrm{sr}}(t, y) = \max\{0, \min\{1, (1 - ty)/2\}\},$$

instead of the popular *hinge loss* that does not satisfy (2). Let $I(g) = \mathbb{E}_{(X,Y)}[\ell_{01}(g(X), Y)]$ be the risk of $g$ w.r.t. $\ell_{01}$ under $p(x, y)$. Then, $\ell_{\mathrm{sr}}$ is neither an upper bound of $\ell_{01}$ so that $I(g) \le R(g)$ is not guaranteed, nor a convex loss so that it gets more difficult to know whether $\ell_{\mathrm{sr}}$ is *classification-calibrated* or not [19].[1] If it is, we are able to control the *excess risk* w.r.t. $\ell_{01}$ by that w.r.t. $\ell$. Here we prove the classification calibration of $\ell_{\mathrm{sr}}$, and consequently it is a safe surrogate loss for $\ell_{01}$.

**Theorem 1.** *The scaled ramp loss $\ell_{\mathrm{sr}}$ is classification-calibrated* (see Appendix A for the proof).

## 3   Theoretical comparisons based on risk bounds

When learning is involved, suppose we are given a *function class* $\mathcal{G}$, and let $g^* = \arg\min_{g\in\mathcal{G}} R(g)$ be the optimal decision function in $\mathcal{G}$, $\hat{g}_{\mathrm{pn}} = \arg\min_{g\in\mathcal{G}} \widehat{R}_{\mathrm{pn}}(g)$, $\hat{g}_{\mathrm{pu}} = \arg\min_{g\in\mathcal{G}} \widehat{R}_{\mathrm{pu}}(g)$, and $\hat{g}_{\mathrm{nu}} = \arg\min_{g\in\mathcal{G}} \widehat{R}_{\mathrm{nu}}(g)$ be arbitrary global minimizers to three risk estimators. Furthermore, let $R^* = \inf_g R(g)$ and $I^* = \inf_g I(g)$ denote the Bayes risks w.r.t. $\ell$ and $\ell_{01}$, where the infimum of $g$ is over all measurable functions.

In this section, we derive and compare risk bounds of three risk minimizers $\hat{g}_{\mathrm{pn}}$, $\hat{g}_{\mathrm{pu}}$ and $\hat{g}_{\mathrm{nu}}$ under the following mild assumption on $\mathcal{G}$, $p(x)$, $p_+(x)$ and $p_-(x)$: There is a constant $C_\mathcal{G} > 0$ such that

$$\mathfrak{R}_{n,q}(\mathcal{G}) \le C_\mathcal{G}/\sqrt{n} \tag{5}$$

for any marginal density $q(x) \in \{p(x), p_+(x), p_-(x)\}$, where

$$\mathfrak{R}_{n,q}(\mathcal{G}) = \mathbb{E}_{\mathcal{X}\sim q^n}\mathbb{E}_\sigma\left[\sup_{g\in\mathcal{G}} \tfrac{1}{n}\sum_{x_i\in\mathcal{X}}\sigma_i g(x_i)\right]$$

is the *Rademacher complexity* of $\mathcal{G}$ for the sampling of size $n$ from $q(x)$ (that is, $\mathcal{X} = \{x_1,\ldots,x_n\}$ and $\sigma = \{\sigma_1,\ldots,\sigma_n\}$, with each $x_i$ drawn from $q(x)$ and each $\sigma_i$ as a *Rademacher variable*) [18]. A special case is covered, namely, sets of *hyperplanes with bounded normals and feature maps*:

$$\mathcal{G} = \{g(x) = \langle w, \phi(x)\rangle_\mathcal{H} \mid \|w\|_\mathcal{H} \le C_w, \|\phi(x)\|_\mathcal{H} \le C_\phi\}, \tag{6}$$

where $\mathcal{H}$ is a Hilbert space with an inner product $\langle\cdot,\cdot\rangle_\mathcal{H}$, $w \in \mathcal{H}$ is a normal vector, $\phi : \mathbb{R}^d \to \mathcal{H}$ is a feature map, and $C_w > 0$ and $C_\phi > 0$ are constants [26].

### 3.1 Risk bounds

Let $L_\ell$ be the *Lipschitz constant* of $\ell$ in its first parameter. To begin with, we establish the learning guarantee of $\hat{g}_{\mathrm{pu}}$ (the proof can be found in Appendix A).

**Theorem 2.** *Assume* (2). *For any* $\delta > 0$, *with probability at least* $1 - \delta$,[2]

$$R(\hat{g}_{\mathrm{pu}}) - R(g^*) \leq 8\pi L_\ell \mathfrak{R}_{n_+,p_+}(\mathcal{G}) + 4L_\ell \mathfrak{R}_{n_{\mathrm{u}},p}(\mathcal{G}) + 2\pi\sqrt{\tfrac{2\ln(4/\delta)}{n_+}} + \sqrt{\tfrac{2\ln(4/\delta)}{n_{\mathrm{u}}}}, \qquad (7)$$

*where* $\mathfrak{R}_{n_+,p_+}(\mathcal{G})$ *and* $\mathfrak{R}_{n_{\mathrm{u}},p}(\mathcal{G})$ *are the Rademacher complexities of* $\mathcal{G}$ *for the sampling of size* $n_+$ *from* $p_+(x)$ *and the sampling of size* $n_{\mathrm{u}}$ *from* $p(x)$. *Moreover, if* $\ell$ *is a classification-calibrated loss, there exists nondecreasing* $\varphi$ *with* $\varphi(0) = 0$, *such that with probability at least* $1 - \delta$,

$$I(\hat{g}_{\mathrm{pu}}) - I^* \leq \varphi\Big( R(g^*) - R^* + 8\pi L_\ell \mathfrak{R}_{n_+,p_+}(\mathcal{G}) + 4L_\ell \mathfrak{R}_{n_{\mathrm{u}},p}(\mathcal{G}) + 2\pi\sqrt{\tfrac{2\ln(4/\delta)}{n_+}} + \sqrt{\tfrac{2\ln(4/\delta)}{n_{\mathrm{u}}}} \Big). \quad (8)$$

In Theorem 2, $R(\hat{g}_{\mathrm{pu}})$ and $I(\hat{g}_{\mathrm{pu}})$ are w.r.t. $p(x,y)$, though $\hat{g}_{\mathrm{pu}}$ is trained from two samples following $p_+(x)$ and $p(x)$. We can see that (7) is an upper bound of the *estimation error* of $\hat{g}_{\mathrm{pu}}$ w.r.t. $\ell$, whose right-hand side (RHS) is small if $\mathcal{G}$ is small; (8) is an upper bound of the *excess risk* of $\hat{g}_{\mathrm{pu}}$ w.r.t. $\ell_{01}$, whose RHS also involves the *approximation error* of $\mathcal{G}$ (i.e., $R(g^*) - R^*$) that is small if $\mathcal{G}$ is large. When $\mathcal{G}$ is fixed and satisfies (5), we have $\mathfrak{R}_{n_+,p_+}(\mathcal{G}) = \mathcal{O}(1/\sqrt{n_+})$ and $\mathfrak{R}_{n_{\mathrm{u}},p}(\mathcal{G}) = \mathcal{O}(1/\sqrt{n_{\mathrm{u}}})$, and then

$$R(\hat{g}_{\mathrm{pu}}) - R(g^*) \to 0, \quad I(\hat{g}_{\mathrm{pu}}) - I^* \to \varphi(R(g^*) - R^*)$$

in $\mathcal{O}_p(1/\sqrt{n_+} + 1/\sqrt{n_{\mathrm{u}}})$. On the other hand, when the size of $\mathcal{G}$ grows with $n_+$ and $n_{\mathrm{u}}$ properly, those complexities of $\mathcal{G}$ vanish slower in order than $\mathcal{O}(1/\sqrt{n_+})$ and $\mathcal{O}(1/\sqrt{n_{\mathrm{u}}})$ but we may have

$$R(\hat{g}_{\mathrm{pu}}) - R(g^*) \to 0, \quad I(\hat{g}_{\mathrm{pu}}) - I^* \to 0,$$

which means $\hat{g}_{\mathrm{pu}}$ approaches the Bayes classifier if $\ell$ is a classification-calibrated loss, in an order slower than $\mathcal{O}_p(1/\sqrt{n_+} + 1/\sqrt{n_{\mathrm{u}}})$ due to the growth of $\mathcal{G}$.

Similarly, we can derive the learning guarantees of $\hat{g}_{\mathrm{pn}}$ and $\hat{g}_{\mathrm{nu}}$ for comparisons. We will just focus on estimation error bounds, because excess risk bounds are their immediate corollaries.

**Theorem 3.** *Assume* (2). *For any* $\delta > 0$, *with probability at least* $1 - \delta$,

$$R(\hat{g}_{\mathrm{pn}}) - R(g^*) \leq 4\pi L_\ell \mathfrak{R}_{n_+,p_+}(\mathcal{G}) + 4(1-\pi)L_\ell \mathfrak{R}_{n_-,p_-}(\mathcal{G}) + \pi\sqrt{\tfrac{2\ln(4/\delta)}{n_+}} + (1-\pi)\sqrt{\tfrac{2\ln(4/\delta)}{n_-}}, \tag{9}$$

*where* $\mathfrak{R}_{n_-,p_-}(\mathcal{G})$ *is the Rademacher complexity of* $\mathcal{G}$ *for the sampling of size* $n_-$ *from* $p_-(x)$.

**Theorem 4.** *Assume* (2). *For any* $\delta > 0$, *with probability at least* $1 - \delta$,

$$R(\hat{g}_{\mathrm{nu}}) - R(g^*) \leq 4L_\ell \mathfrak{R}_{n_{\mathrm{u}},p}(\mathcal{G}) + 8(1-\pi)L_\ell \mathfrak{R}_{n_-,p_-}(\mathcal{G}) + \sqrt{\tfrac{2\ln(4/\delta)}{n_{\mathrm{u}}}} + 2(1-\pi)\sqrt{\tfrac{2\ln(4/\delta)}{n_-}}. \quad (10)$$

In order to compare the bounds, we simplify (9), (7) and (10) using Eq. (5). To this end, we define $f(\delta) = 4L_\ell C_\mathcal{G} + \sqrt{2\ln(4/\delta)}$. For the special case of $\mathcal{G}$ defined in (6), define $f(\delta)$ accordingly as $f(\delta) = 4L_\ell C_w C_\phi + \sqrt{2\ln(4/\delta)}$.

**Corollary 5.** *The estimation error bounds below hold separately with probability at least* $1 - \delta$:

$$R(\hat{g}_{\mathrm{pn}}) - R(g^*) \leq f(\delta) \cdot \{\pi/\sqrt{n_+} + (1-\pi)/\sqrt{n_-}\}, \tag{11}$$

$$R(\hat{g}_{\mathrm{pu}}) - R(g^*) \leq f(\delta) \cdot \{2\pi/\sqrt{n_+} + 1/\sqrt{n_{\mathrm{u}}}\}, \tag{12}$$

$$R(\hat{g}_{\mathrm{nu}}) - R(g^*) \leq f(\delta) \cdot \{1/\sqrt{n_{\mathrm{u}}} + 2(1-\pi)/\sqrt{n_-}\}. \tag{13}$$

### 3.2 Finite-sample comparisons

Note that three risk minimizers $\hat{g}_{\mathrm{pn}}$, $\hat{g}_{\mathrm{pu}}$ and $\hat{g}_{\mathrm{nu}}$ work in similar problem settings and their bounds in Corollary 5 are proven using exactly the same proof technique. Then, the differences in bounds reflect the intrinsic differences between risk minimizers. Let us compare those bounds. Define

$$\alpha_{\mathrm{pu,pn}} = \big(\pi/\sqrt{n_+} + 1/\sqrt{n_{\mathrm{u}}}\big) \big/ \big((1-\pi)/\sqrt{n_-}\big), \tag{14}$$

$$\alpha_{\mathrm{nu,pn}} = \big((1-\pi)/\sqrt{n_-} + 1/\sqrt{n_{\mathrm{u}}}\big) \big/ \big(\pi/\sqrt{n_+}\big). \tag{15}$$

Eqs. (14) and (15) constitute our first main result.

Table 1: Properties of $\alpha_{\mathrm{pu,pn}}$ and $\alpha_{\mathrm{nu,pn}}$.

| | no specification | | sizes are proportional | | $\rho_{\mathrm{pn}} = \pi/(1-\pi)$ | |
|---|---|---|---|---|---|---|
| | mono. inc. | mono. dec. | mono. inc. | mono. dec. | mono. inc. | minimum |
| $\alpha_{\mathrm{pu,pn}}$ | $\pi, n_-$ | $n_+, n_{\mathrm{u}}$ | $\pi, \rho_{\mathrm{pu}}$ | $\rho_{\mathrm{pn}}$ | $\rho_{\mathrm{pu}}$ | $2\sqrt{\rho_{\mathrm{pu}} + \sqrt{\rho_{\mathrm{pu}}}}$ |
| $\alpha_{\mathrm{nu,pn}}$ | $n_+$ | $\pi, n_-, n_{\mathrm{u}}$ | $\rho_{\mathrm{pn}}, \rho_{\mathrm{nu}}$ | $\pi$ | $\rho_{\mathrm{nu}}$ | $2\sqrt{\rho_{\mathrm{nu}} + \sqrt{\rho_{\mathrm{nu}}}}$ |

**Theorem 6** (Finite-sample comparisons). *Assume* (5) *is satisfied. Then the estimation error bound of $\hat{g}_{\mathrm{pu}}$ in* (12) *is tighter than that of $\hat{g}_{\mathrm{pn}}$ in* (11) *if and only if $\alpha_{\mathrm{pu,pn}} < 1$; also, the estimation error bound of $\hat{g}_{\mathrm{nu}}$ in* (13) *is tighter than that of $\hat{g}_{\mathrm{pn}}$ if and only if $\alpha_{\mathrm{nu,pn}} < 1$.*

*Proof.* Fix $\pi$, $n_+$, $n_-$ and $n_{\mathrm{u}}$, and then denote by $V_{\mathrm{pn}}$, $V_{\mathrm{pu}}$ and $V_{\mathrm{nu}}$ the values of the RHSs of (11), (12) and (13). In fact, the definitions of $\alpha_{\mathrm{pu,pn}}$ and $\alpha_{\mathrm{nu,pn}}$ in (14) and (15) came from

$$\alpha_{\mathrm{pu,pn}} = \frac{V_{\mathrm{pu}} - \pi f(\delta)/\sqrt{n_+}}{V_{\mathrm{pn}} - \pi f(\delta)/\sqrt{n_+}}, \quad \alpha_{\mathrm{nu,pn}} = \frac{V_{\mathrm{nu}} - (1-\pi)f(\delta)/\sqrt{n_-}}{V_{\mathrm{pn}} - (1-\pi)f(\delta)/\sqrt{n_-}}.$$

As a consequence, compared with $V_{\mathrm{pn}}$, $V_{\mathrm{pu}}$ is smaller and (12) is tighter if and only if $\alpha_{\mathrm{pu,pn}} < 1$, and $V_{\mathrm{nu}}$ is smaller and (13) is tighter if and only if $\alpha_{\mathrm{nu,pn}} < 1$. $\square$

We analyze some properties of $\alpha_{\mathrm{pu,pn}}$ before going to our second main result. The most important property is that it relies on $\pi$, $n_+$, $n_-$ and $n_{\mathrm{u}}$ only; it is independent of $\mathcal{G}$, $p(x,y)$, $p(x)$, $p_+(x)$ and $p_-(x)$ as long as (5) is satisfied. Next, $\alpha_{\mathrm{pu,pn}}$ is obviously a monotonic function of $\pi$, $n_+$, $n_-$ and $n_{\mathrm{u}}$. Furthermore, it is unbounded no matter if $\pi$ is fixed or not. Properties of $\alpha_{\mathrm{nu,pn}}$ are similar, as summarized in Table 1.

Implications of the monotonicity of $\alpha_{\mathrm{pu,pn}}$ are given as follows. Intuitively, when other factors are fixed, larger $n_{\mathrm{u}}$ or $n_-$ improves $\hat{g}_{\mathrm{pu}}$ or $\hat{g}_{\mathrm{pn}}$ respectively. However, it is complicated why $\alpha_{\mathrm{pu,pn}}$ is monotonically decreasing with $n_+$ and increasing with $\pi$. The weights of the empirical average of $\mathcal{X}_+$ is $2\pi$ in $\widehat{R}_{\mathrm{pu}}(g)$ and $\pi$ in $\widehat{R}_{\mathrm{pn}}(g)$, as in $\widehat{R}_{\mathrm{pu}}(g)$ it also joins the estimation of $(1-\pi)R_-(g)$. It makes $\mathcal{X}_+$ more important for $\widehat{R}_{\mathrm{pu}}(g)$, and thus larger $n_+$ improves $\hat{g}_{\mathrm{pu}}$ more than $\hat{g}_{\mathrm{pn}}$. Moreover, $(1-\pi)R_-(g)$ is directly estimated in $\widehat{R}_{\mathrm{pn}}(g)$ and the concentration $\mathcal{O}_p((1-\pi)/\sqrt{n_-})$ is better if $\pi$ is larger, whereas it is indirectly estimated through $R_{\mathrm{u},-}(g) - \pi(1 - R_+(g))$ in $\widehat{R}_{\mathrm{pu}}(g)$ and the concentration $\mathcal{O}_p(\pi/\sqrt{n_+} + 1/\sqrt{n_{\mathrm{u}}})$ is worse if $\pi$ is larger. As a result, when the sample sizes are fixed $\hat{g}_{\mathrm{pu}}$ is more (or less) favorable as $\pi$ decreases (or increases).

A natural question is what the monotonicity of $\alpha_{\mathrm{pu,pn}}$ would be if we enforce $n_+$, $n_-$ and $n_{\mathrm{u}}$ to be proportional. To answer this question, we assume $n_+/n_- = \rho_{\mathrm{pn}}$, $n_+/n_{\mathrm{u}} = \rho_{\mathrm{pu}}$ and $n_-/n_{\mathrm{u}} = \rho_{\mathrm{nu}}$ where $\rho_{\mathrm{pn}}$, $\rho_{\mathrm{pu}}$ and $\rho_{\mathrm{nu}}$ are certain constants, then (14) and (15) can be rewritten as

$$\alpha_{\mathrm{pu,pn}} = (\pi + \sqrt{\rho_{\mathrm{pu}}})/((1-\pi)\sqrt{\rho_{\mathrm{pn}}}), \quad \alpha_{\mathrm{nu,pn}} = (1 - \pi + \sqrt{\rho_{\mathrm{nu}}})/(\pi/\sqrt{\rho_{\mathrm{pn}}}).$$

As shown in Table 1, $\alpha_{\mathrm{pu,pn}}$ is now increasing with $\rho_{\mathrm{pu}}$ and decreasing with $\rho_{\mathrm{pn}}$. It is because, for instance, when $\rho_{\mathrm{pn}}$ is fixed and $\rho_{\mathrm{pu}}$ increases, $n_{\mathrm{u}}$ is meant to decrease relatively to $n_+$ and $n_-$.

Finally, the properties will dramatically change if we enforce $\rho_{\mathrm{pn}} = \pi/(1-\pi)$ that approximately holds in ordinary supervised learning. Under this constraint, we have

$$\alpha_{\mathrm{pu,pn}} = (\pi + \sqrt{\rho_{\mathrm{pu}}})/\sqrt{\pi(1-\pi)} \geq 2\sqrt{\rho_{\mathrm{pu}} + \sqrt{\rho_{\mathrm{pu}}}},$$

where the equality is achieved at $\bar{\pi} = \sqrt{\rho_{\mathrm{pu}}}/(2\sqrt{\rho_{\mathrm{pu}}} + 1)$. Here, $\alpha_{\mathrm{pu,pn}}$ decreases with $\pi$ if $\pi < \bar{\pi}$ and increases with $\pi$ if $\pi > \bar{\pi}$, though it is not convex in $\pi$. Only if $n_{\mathrm{u}}$ is sufficiently larger than $n_+$ (e.g., $\rho_{\mathrm{pu}} < 0.04$), could $\alpha_{\mathrm{pu,pn}} < 1$ be possible and $\hat{g}_{\mathrm{pu}}$ have a tighter estimation error bound.

### 3.3 Asymptotic comparisons

In practice, we may find that $\hat{g}_{\mathrm{pu}}$ is worse than $\hat{g}_{\mathrm{pn}}$ and $\alpha_{\mathrm{pu,pn}} > 1$ given $\mathcal{X}_+$, $\mathcal{X}_-$ and $\mathcal{X}_{\mathrm{u}}$. This is probably the consequence especially when $n_{\mathrm{u}}$ is not sufficiently larger than $n_+$ and $n_-$. Should we then try to collect much more U data or just give up PU learning? Moreover, if we are able to have as many U data as possible, is there any solution that would be provably better than PN learning?

We answer these questions by asymptotic comparisons. Notice that each pair of $(n_+, n_\mathrm{u})$ yields a value of the RHS of (12), each $(n_+, n_-)$ yields a value of the RHS of (11), and consequently each triple of $(n_+, n_-, n_\mathrm{u})$ determines a value of $\alpha_\mathrm{pu,pn}$. Define the limits of $\alpha_\mathrm{pu,pn}$ and $\alpha_\mathrm{nu,pn}$ as

$$\alpha_\mathrm{pu,pn}^* = \lim_{n_+,n_-,n_\mathrm{u}\to\infty} \alpha_\mathrm{pu,pn}, \quad \alpha_\mathrm{nu,pn}^* = \lim_{n_+,n_-,n_\mathrm{u}\to\infty} \alpha_\mathrm{nu,pn}.$$

Recall that $n_+$, $n_-$ and $n_\mathrm{u}$ are independent, and we need two conditions for the existence of $\alpha_\mathrm{pu,pn}^*$ and $\alpha_\mathrm{nu,pn}^*$: $n_+ \to \infty$ *and* $n_- \to \infty$ *in the same order* and $n_\mathrm{u} \to \infty$ *faster in order than them*. It is a bit stricter than what is necessary, but is consistent with a practical assumption: *P and N data are roughly equally expensive, whereas U data are much cheaper than P and N data*. Intuitively, since $\alpha_\mathrm{pu,pn}$ and $\alpha_\mathrm{nu,pn}$ measure relative qualities of the estimation error bounds of $\hat{g}_\mathrm{pu}$ and $\hat{g}_\mathrm{nu}$ against that of $\hat{g}_\mathrm{pn}$, $\alpha_\mathrm{pu,pn}^*$ and $\alpha_\mathrm{nu,pn}^*$ measure relative qualities of the *limits of those bounds* accordingly.

In order to illustrate properties of $\alpha_\mathrm{pu,pn}^*$ and $\alpha_\mathrm{nu,pn}^*$, assume only $n_\mathrm{u}$ approaches infinity while $n_+$ and $n_-$ stay finite, so that $\alpha_\mathrm{pu,pn}^* = \pi\sqrt{n_-}/((1-\pi)\sqrt{n_+})$ and $\alpha_\mathrm{nu,pn}^* = (1-\pi)\sqrt{n_+}/(\pi\sqrt{n_-})$. Thus, $\alpha_\mathrm{pu,pn}^*\alpha_\mathrm{nu,pn}^* = 1$, which implies $\alpha_\mathrm{pu,pn}^* < 1$ or $\alpha_\mathrm{nu,pn}^* < 1$ unless $n_+/n_- = \pi^2/(1-\pi)^2$. In principle, this exception should be exceptionally rare since $n_+/n_-$ is a rational number whereas $\pi^2/(1-\pi)^2$ is a real number. This argument constitutes our second main result.

**Theorem 7** (Asymptotic comparisons). *Assume* (5) *and one set of conditions below are satisfied:*

    *(a)* $n_+ < \infty$, $n_- < \infty$ *and* $n_\mathrm{u} \to \infty$. *In this case, let* $\alpha^* = (\pi\sqrt{n_-})/((1-\pi)\sqrt{n_+})$;

    *(b)* $0 < \lim_{n_+,n_-\to\infty} n_+/n_- < \infty$ *and* $\lim_{n_+,n_-,n_\mathrm{u}\to\infty}(n_+ + n_-)/n_\mathrm{u} = 0$. *In this case, let* $\alpha^* = \pi/((1-\pi)\sqrt{\rho_\mathrm{pn}^*})$ *where* $\rho_\mathrm{pn}^* = \lim_{n_+,n_-\to\infty} n_+/n_-$.

*Then, either the limit of estimation error bounds of $\hat{g}_\mathrm{pu}$ will improve on that of $\hat{g}_\mathrm{pn}$ (i.e., $\alpha_\mathrm{pu,pn}^* < 1$) if $\alpha^* < 1$, or the limit of bounds of $\hat{g}_\mathrm{nu}$ will improve on that of $\hat{g}_\mathrm{pn}$ (i.e., $\alpha_\mathrm{nu,pn}^* < 1$) if $\alpha^* > 1$. The only exception is $n_+/n_- = \pi^2/(1-\pi)^2$ in (a) or $\rho_\mathrm{pn}^* = \pi^2/(1-\pi)^2$ in (b).*

*Proof.* Note that $\alpha^* = \alpha_\mathrm{pu,pn}^*$ in both cases. The proof of case (a) has been given as an illustration of the properties of $\alpha_\mathrm{pu,pn}^*$ and $\alpha_\mathrm{nu,pn}^*$. The proof of case (b) is analogous. □

As a result, when we find that $\hat{g}_\mathrm{pu}$ is worse than $\hat{g}_\mathrm{pn}$ and $\alpha_\mathrm{pu,pn} > 1$, we should look at $\alpha^*$ defined in Theorem 7. If $\alpha^* < 1$, $\hat{g}_\mathrm{pu}$ is promising and we should collect more U data; if $\alpha^* > 1$ otherwise, we should give up $\hat{g}_\mathrm{pu}$, but instead $\hat{g}_\mathrm{nu}$ is promising and we should collect more U data as well. In addition, the gap between $\alpha^*$ and one indicates how many U data would be sufficient. If the gap is significant, slightly more U data may be enough; if the gap is slight, significantly more U data may be necessary. In practice, however, U data are cheaper but not free, and we cannot have as many U data as possible. Therefore, $\hat{g}_\mathrm{pn}$ is still of practical importance given limited budgets.

## 3.4 Remarks

Theorem 2 relies on a fundamental lemma of the *uniform deviation* from the risk estimator $\widehat{R}_\mathrm{pu}(g)$ to the risk $R(g)$:

**Lemma 8.** *For any $\delta > 0$, with probability at least $1 - \delta$,*

$$\sup_{g\in\mathcal{G}} |\widehat{R}_\mathrm{pu}(g) - R(g)| \le 4\pi L_\ell \mathfrak{R}_{n_+,p_+}(\mathcal{G}) + 2L_\ell \mathfrak{R}_{n_\mathrm{u},p}(\mathcal{G}) + 2\pi\sqrt{\frac{\ln(4/\delta)}{2n_+}} + \sqrt{\frac{\ln(4/\delta)}{2n_\mathrm{u}}}.$$

In Lemma 8, $R(g)$ is w.r.t. $p(x,y)$, though $\widehat{R}_\mathrm{pu}(g)$ is w.r.t. $p_+(x)$ and $p(x)$. Rademacher complexities are also w.r.t. $p_+(x)$ and $p(x)$, and they can be bounded easily for $\mathcal{G}$ defined in Eq. (6).

Theorems 6 and 7 rely on (5). Thanks to it, we can simplify Theorems 2, 3 and 4. In fact, (5) holds for not only the special case of $\mathcal{G}$ defined in (6), but also the vast majority of discriminative models in machine learning that are nonlinear in parameters such as *decision trees* (cf. Theorem 17 in [16]) and *feedforward neural networks* (cf. Theorem 18 in [16]).

Theorem 2 in [7] is a similar bound of the same order as our Lemma 8. That theorem is based on a tricky decomposition of the risk

$$\mathbb{E}_{(X,Y)}[\ell(g(X),Y)] = \pi\mathbb{E}_+[\tilde{\ell}(g(X),+1)] + \mathbb{E}_{(X,Y)}[\tilde{\ell}(g(X),Y)],$$

where the surrogate loss $\tilde{\ell}(t,y) = (2/(y+3))\ell(t,y)$ is not $\ell$ for risk minimization and labels of $\mathcal{X}_\mathrm{u}$ are needed for risk evaluation, so that no further bound is implied. Lemma 8 uses the same $\ell$ as risk minimization and requires no label of $\mathcal{X}_\mathrm{u}$ for evaluating $\widehat{R}_\mathrm{pu}(g)$, so that it can serve as the stepping stone to our estimation error bound in Theorem 2.

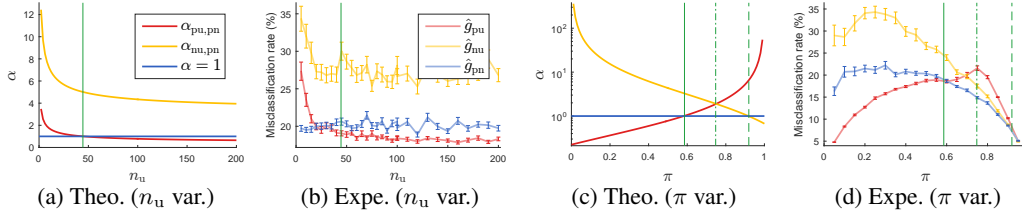

|(a) Theo. ($n_\mathrm{u}$ var.)|(b) Expe. ($n_\mathrm{u}$ var.)|(c) Theo. ($\pi$ var.)|(d) Expe. ($\pi$ var.)|

Figure 1: Theoretical and experimental results based on artificial data.

## 4 Experiments

In this section, we experimentally validate our theoretical findings.

**Artificial data**  Here, $\mathcal{X}_+$, $\mathcal{X}_-$ and $\mathcal{X}_\mathrm{u}$ are in $\mathbb{R}^2$ and drawn from three marginal densities

$$p_+(x) = N(+1_2/\sqrt{2}, I_2), \quad p_-(x) = N(-1_2/\sqrt{2}, I_2), \quad p(x) = \pi p_+(x) + (1-\pi)p_-(x),$$

where $N(\mu, \Sigma)$ is the normal distribution with mean $\mu$ and covariance $\Sigma$, $1_2$ and $I_2$ are the all-one vector and identity matrix of size 2. The test set contains one million data drawn from $p(x, y)$.

The model $g(x) = \langle w, x \rangle + b$ where $w \in \mathbb{R}^2, b \in \mathbb{R}$ and the scaled ramp loss $\ell_\mathrm{sr}$ are employed. In addition, an $\ell_2$-regularization is added with the regularization parameter fixed to $10^{-3}$, and there is no hard constraint on $\|w\|_2$ or $\|x\|_2$ as in Eq. (6). The solver for minimizing three regularized risk estimators comes from [7] (refer also to [27, 28] for the optimization technique).

The results are reported in Figure 1. In (a)(b), $n_+ = 45$, $n_- = 5$, $\pi = 0.5$, and $n_\mathrm{u}$ varies from 5 to 200; in (c)(d), $n_+ = 45$, $n_- = 5$, $n_\mathrm{u} = 100$, and $\pi$ varies from 0.05 to 0.95. Specifically, (a) shows $\alpha_\mathrm{pu,pn}$ and $\alpha_\mathrm{nu,pn}$ as functions of $n_\mathrm{u}$, and (c) shows them as functions of $\pi$. For the experimental results, $\hat{g}_\mathrm{pn}$, $\hat{g}_\mathrm{pu}$ and $\hat{g}_\mathrm{nu}$ were trained based on 100 random samplings for every $n_\mathrm{u}$ in (b) and $\pi$ in (d), and means with standard errors of the misclassification rates are shown, as $\ell_\mathrm{sr}$ is classification-calibrated. Note that the empirical misclassification rates are essentially the risks w.r.t. $\ell_{01}$ as there were one million test data, and the fluctuations are attributed to the non-convex nature of $\ell_\mathrm{sr}$. Also, the curve of $\hat{g}_\mathrm{pn}$ is not a flat line in (b), since its training data at every $n_\mathrm{u}$ were exactly same as the training data of $\hat{g}_\mathrm{pu}$ and $\hat{g}_\mathrm{nu}$ for fair experimental comparisons.

In Figure 1, the theoretical and experimental results are highly consistent. The red and blue curves intersect at nearly the same positions in (a)(b) and in (c)(d), even though the risk minimizers in the experiments were locally optimal and regularized, making our estimation error bounds inexact.

**Benchmark data**  Table 2 summarizes the specification of benchmarks, which were downloaded from many sources including the *IDA benchmark repository* [29], the *UCI machine learning repository*, the *semi-supervised learning book* [30], and the *European ESPRIT 5516 project*.[3] In Table 2, three rows describe the number of features, the number of data, and the ratio of P data according to the true class labels. Given a random sampling of $\mathcal{X}_+$, $\mathcal{X}_-$ and $\mathcal{X}_\mathrm{u}$, the test set has all the remaining data if they are less than $10^4$, or else drawn uniformly from the remaining data of size $10^4$.

For benchmark data, the linear model for the artificial data is not enough, and its kernel version is employed. Consider training $\hat{g}_\mathrm{pu}$ for example. Given a random sampling, $g(x) = \langle w, \phi(x) \rangle + b$ is used where $w \in \mathbb{R}^{n_+ + n_\mathrm{u}}, b \in \mathbb{R}$ and $\phi: \mathbb{R}^d \to \mathbb{R}^{n_+ + n_\mathrm{u}}$ is the *empirical kernel map* [26] based on $\mathcal{X}_+$ and $\mathcal{X}_\mathrm{u}$ for the *Gaussian kernel*. The kernel width and the regularization parameter are selected by five-fold cross-validation for each risk minimizer and each random sampling.

Table 2: Specification of benchmark datasets.

|         | banana | phoneme | magic | image | german | twonorm | waveform | spambase | coil2 |
|---------|--------|---------|-------|-------|--------|---------|----------|----------|-------|
| dim     | 2      | 5       | 10    | 18    | 20     | 20      | 21       | 57       | 241   |
| size    | 5300   | 5404    | 19020 | 2086  | 1000   | 7400    | 5000     | 4597     | 1500  |
| P ratio | .448   | .293    | .648  | .570  | .300   | .500    | .329     | .394     | .500  |

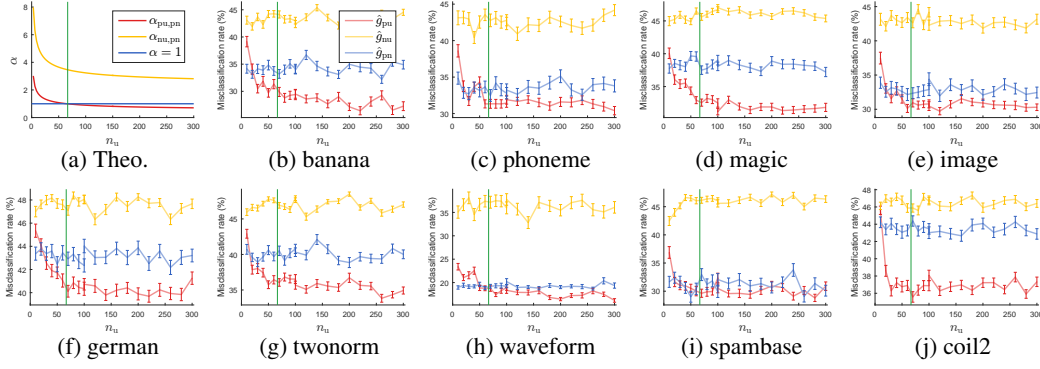

Figure 2: Experimental results based on benchmark data by varying $n_{\mathrm{u}}$.

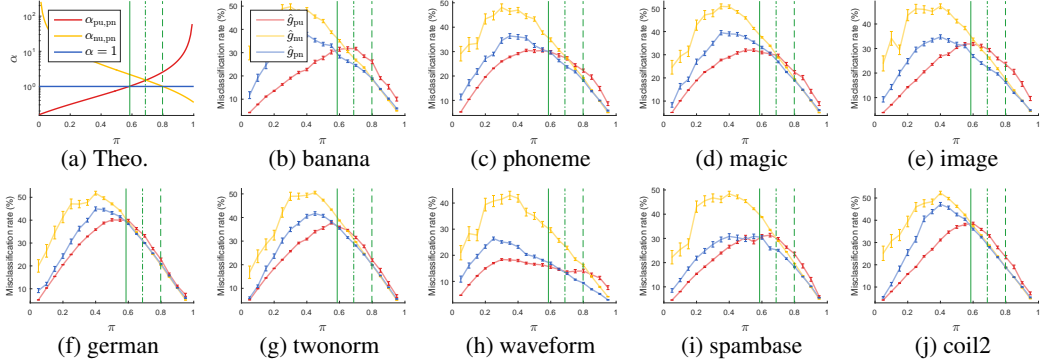

Figure 3: Experimental results based on benchmark data by varying $\pi$.

The results by varying $n_{\mathrm{u}}$ and $\pi$ are reported in Figures 2 and 3 respectively. Similarly to Figure 1, in Figure 2, $n_+ = 25$, $n_- = 5$, $\pi = 0.5$, and $n_{\mathrm{u}}$ varies from 10 to 300, while in Figure 3, $n_+ = 25$, $n_- = 5$, $n_{\mathrm{u}} = 200$, and $\pi$ varies from 0.05 to 0.95. Figures 2(a) and 3(a) depict $\alpha_{\mathrm{pu,pn}}$ and $\alpha_{\mathrm{nu,pn}}$ as functions of $n_{\mathrm{u}}$ and $\pi$, and all the remaining subfigures depict means with standard errors of the misclassification rates based on 100 random samplings for every $n_{\mathrm{u}}$ and $\pi$.

The theoretical and experimental results based on benchmarks are still highly consistent. However, unlike in Figure 1(b), in Figure 2 only the errors of $\hat{g}_{\mathrm{pu}}$ decrease with $n_{\mathrm{u}}$, and the errors of $\hat{g}_{\mathrm{nu}}$ just fluctuate randomly. This may be because benchmark data are more difficult than artificial data and hence $n_- = 5$ is not sufficiently informative for $\hat{g}_{\mathrm{nu}}$ even when $n_{\mathrm{u}} = 300$. On the other hand, we can see that Figures 3(a) and 1(c) look alike, and so do all the remaining subfigures in Figure 3 and Figure 1(d). Nevertheless, three intersections in Figure 3(a) are closer than those in Figure 1(c), as $n_{\mathrm{u}} = 200$ in Figure 3(a) and $n_{\mathrm{u}} = 100$ in Figure 1(c). The three intersections will become a single one if $n_{\mathrm{u}} = \infty$. By observing the experimental results, three curves in Figure 3 are also closer than those in Figure 1(d) when $\pi \geq 0.6$, which demonstrates the validity of our theoretical findings.

## 5 Conclusions

In this paper, we studied a fundamental problem in PU learning, namely, when PU learning is likely to outperform PN learning. Estimation error bounds of the risk minimizers were established in PN, PU and NU learning. We found that under the very mild assumption (5): The PU (or NU) bound is tighter than the PN bound, if $\alpha_{\mathrm{pu,pn}}$ in (14) (or $\alpha_{\mathrm{nu,pn}}$ in (15)) is smaller than one (cf. Theorem 6); either the limit of $\alpha_{\mathrm{pu,pn}}$ or that of $\alpha_{\mathrm{nu,pn}}$ will be smaller than one, if the size of U data increases faster in order than the sizes of P and N data (cf. Theorem 7). We validated our theoretical findings experimentally using one artificial data and nine benchmark data.

### Acknowledgments

GN was supported by the JST CREST program and Microsoft Research Asia. MCdP, YM, and MS were supported by the JST CREST program. TS was supported by JSPS KAKENHI 15J09111.

## Footnotes

[1] A loss function $\ell$ is classification-calibrated if and only if there is a convex, invertible and nondecreasing transformation $\psi_\ell$ with $\psi_\ell(0) = 0$, such that $\psi_\ell(I(g) - \inf_g I(g)) \le R(g) - \inf_g R(g)$ [19].

[2]Here, the probability is over repeated sampling of data for training $\hat{g}_{\mathrm{pu}}$, while in Lemma 8, it will be for evaluating $\widehat{R}_{\mathrm{pu}}(g)$.

[3]See http://www.raetschlab.org/Members/raetsch/benchmark/ for IDA, http://archive.ics.uci.edu/ml/ for UCI, http://olivier.chapelle.cc/ssl-book/ for the SSL book and https://www.elen.ucl.ac.be/neural-nets/Research/Projects/ELENA/ for the ELENA project.

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
