[Supplementary Material]

# A  Proofs

In this appendix, we prove Theorem 1 in Section 2, and Lemma 8, Theorem 2, and Corollary 5 in Section 3. The proofs of Theorems 3 and 4 are omitted, since they are essentially similar to that of Theorem 2 relying on slightly different uniform deviation bounds.

## A.1  Proof of Theorem 1

The proof is straightforward. Denote by

$$\pi_+(x) = p(Y = +1 \mid X = x), \quad \pi_-(x) = p(Y = -1 \mid X = x),$$

then the *conditional risk* is

$$\mathbb{E}_Y[\ell_{\mathrm{sr}}(g(X), Y) \mid X = x] = \pi_+(x)\ell_{\mathrm{sr}}(g(x), +1) + \pi_-(x)\ell_{\mathrm{sr}}(g(x), -1)$$

$$= \begin{cases} \pi_+(x), & g(x) \leq -1, \\ 1/2 - (\pi_+(x) - \pi_-(x))g(x)/2, & -1 < g(x) < +1, \\ \pi_-(x), & g(x) \geq +1. \end{cases}$$

The minimum is achieved by $g(x) = \mathrm{sign}(\pi_+(x) - \pi_-(x))$, which is actually the Bayes classifier. Therefore, $\ell_{\mathrm{sr}}$ is classification-calibrated according to Theorem 1.3.c in [19]. $\qquad\square$

## A.2  Proof of Lemma 8

Similarly to the decomposition in Eq. (3) such that

$$R(g) = 2\pi R_+(g) + R_{\mathrm{u},-}(g) - \pi,$$

we have seen in the definition of $\widehat{R}_{\mathrm{pu}}(g)$ that it can also be decomposed into

$$\widehat{R}_{\mathrm{pu}}(g) = 2\pi \widehat{R}_+(g) + \widehat{R}_{\mathrm{u},-}(g) - \pi,$$

where

$$\widehat{R}_+(g) = \frac{1}{n_+}\sum_{x_i \in \mathcal{X}_+} \ell(g(x_i), +1), \quad \widehat{R}_{\mathrm{u},-}(g) = \frac{1}{n_{\mathrm{u}}}\sum_{x_j \in \mathcal{X}_{\mathrm{u}}} \ell(g(x_j), -1)$$

are the empirical averages corresponding to $R_+(g)$ and $R_{\mathrm{u},-}(g)$. Due to the sub-additivity of the supremum operators, it holds that

$$\sup_{g \in \mathcal{G}} |\widehat{R}_{\mathrm{pu}}(g) - R(g)| \leq 2\pi \sup_{g \in \mathcal{G}} |\widehat{R}_+(g) - R_+(g)| + \sup_{g \in \mathcal{G}} |\widehat{R}_{\mathrm{u},-}(g) - R_{\mathrm{u},-}(g)|.$$

As a result, in order to prove Lemma 8, it suffices to show that with probability at least $1 - \delta/2$, the uniform deviation bounds below hold separately:

$$\sup_{g \in \mathcal{G}} |\widehat{R}_+(g) - R_+(g)| \leq 2L_\ell \mathfrak{R}_{n_+, p_+}(\mathcal{G}) + \sqrt{\frac{\ln(4/\delta)}{2n_+}}, \tag{16}$$

$$\sup_{g \in \mathcal{G}} |\widehat{R}_{\mathrm{u},-}(g) - R_{\mathrm{u},-}(g)| \leq 2L_\ell \mathfrak{R}_{n_{\mathrm{u}}, p}(\mathcal{G}) + \sqrt{\frac{\ln(4/\delta)}{2n_{\mathrm{u}}}}. \tag{17}$$

In the following we prove (16), and then (17) can be proven using the same proof technique.

Since the surrogate loss $\ell$ is bounded by 0 and 1 according to (2), the change of $\widehat{R}_+(g)$ will be no more than $1/n_+$ if some $x_i$ in $\mathcal{X}_+$ is replaced with $x_i'$. Thus *McDiarmid's inequality* [31] implies

$$\Pr\left[|\widehat{R}_+(g) - R_+(g)| \geq \epsilon\right] \leq 2\exp\left(-\frac{2\epsilon^2}{n_+(1/n_+)^2}\right)$$

for any fixed $g$. Equivalently, for any fixed $g$, with probability at least $1 - \delta/2$,

$$|\widehat{R}_+(g) - R_+(g)| \leq \sqrt{\frac{\ln(4/\delta)}{2n_+}}.$$

Then, according to the *basic uniform deviation bound* using the Rademacher complexity [18], with probability at least $1 - \delta/2$,

$$\sup_{g \in \mathcal{G}} |\widehat{R}_+(g) - R_+(g)| \leq 2\mathfrak{R}_{n_+, p_+}(\ell \circ \mathcal{G}) + \sqrt{\frac{\ln(4/\delta)}{2n_+}}, \tag{18}$$

where $\mathfrak{R}_{n_+, p_+}(\ell \circ \mathcal{G})$ is the Rademacher complexity of the *composite function class* $(\ell \circ \mathcal{G})$ for the sampling of size $n_+$ from $p_+(x)$ defined by

$$\mathfrak{R}_{n_+, p_+}(\ell \circ \mathcal{G}) = \mathbb{E}_{\mathcal{X}_+ \sim p_+^{n_+}} \mathbb{E}_\sigma \left[\sup_{g \in \mathcal{G}} \frac{1}{n_+}\sum_{x_i \in \mathcal{X}_+} \sigma_i \ell(g(x_i), +1)\right].$$

As $\ell(t, y)$ is $L_\ell$-Lipschitz-continuous in $t$ for every $y$, we have $\mathfrak{R}_{n_+, p_+}(\ell \circ \mathcal{G}) \leq L_\ell \mathfrak{R}_{n_+, p_+}(\mathcal{G})$ by *Talagrand's contraction lemma* [32], which proves (16). $\qquad\square$

### A.3 Proof of Theorem 2

Based on Lemma 8, the estimation error bound (7) is proven through

$$R(\hat{g}_{\mathrm{pu}}) - R(g^*) = \left( \widehat{R}_{\mathrm{pu}}(\hat{g}_{\mathrm{pu}}) - \widehat{R}_{\mathrm{pu}}(g^*) \right) + \left( R(\hat{g}_{\mathrm{pu}}) - \widehat{R}_{\mathrm{pu}}(\hat{g}_{\mathrm{pu}}) \right) + \left( \widehat{R}_{\mathrm{pu}}(g^*) - R(g^*) \right)$$

$$\leq 0 + 2 \sup_{g \in \mathcal{G}} |\widehat{R}_{\mathrm{pu}}(g) - R(g)|$$

$$\leq 8\pi L_\ell \Re_{n_+, p_+}(\mathcal{G}) + 4L_\ell \Re_{n_{\mathrm{u}}, p}(\mathcal{G}) + 2\pi \sqrt{\frac{2\ln(4/\delta)}{n_+}} + \sqrt{\frac{2\ln(4/\delta)}{n_{\mathrm{u}}}},$$

where we have used $\widehat{R}_{\mathrm{pu}}(\hat{g}_{\mathrm{pu}}) \leq \widehat{R}_{\mathrm{pu}}(g^*)$ by the definition of $\hat{g}_{\mathrm{pu}}$.

Moreover, if $\ell$ is classification-calibrated, Theorem 1 in [19] implies that there will exist a convex, invertible and nondecreasing transformation $\psi_\ell$ with $\psi_\ell(0) = 0$, such that

$$\psi_\ell(I(\hat{g}_{\mathrm{pu}}) - I^*) \leq R(\hat{g}_{\mathrm{pu}}) - R^*.$$

Hence, let $\varphi = \psi_\ell^{-1}$, we have

$$I(\hat{g}_{\mathrm{pu}}) - I^* \leq \varphi(R(\hat{g}_{\mathrm{pu}}) - R^*)$$
$$= \varphi(R(g^*) - R^* + R(\hat{g}_{\mathrm{pu}}) - R(g^*)),$$

and subsequently the excess risk bound (8) is an immediate corollary of (7). $\qquad\square$

### A.4 Proof of Corollary 5

Given (5), the estimation error bound (7) can be rewritten into

$$R(\hat{g}_{\mathrm{pu}}) - R(g^*) \leq 8\pi L_\ell C_\mathcal{G}/\sqrt{n_+} + 2\pi \sqrt{\frac{2\ln(4/\delta)}{n_+}} + 4L_\ell C_\mathcal{G}/\sqrt{n_{\mathrm{u}}} + \sqrt{\frac{2\ln(4/\delta)}{n_{\mathrm{u}}}}$$

$$= 2\pi f(\delta)/\sqrt{n_+} + f(\delta)/\sqrt{n_{\mathrm{u}}},$$

where $f(\delta) = 4L_\ell C_\mathcal{G} + \sqrt{2\ln(4/\delta)}$. This proves (12). In exactly the same way, we could get (11) from (9) and (13) from (10).

Consider the special case of $\mathcal{G}$ defined in (6). Recall that $\Re_{n,q}(\mathcal{G})$ is the Rademacher complexity of $\mathcal{G}$ for $\mathcal{X} = \{x_1, \ldots, x_n\}$ with each $x_i$ drawn from $q(x)$. Given any such $\mathcal{X}$, denote by $\widehat{\Re}_{\mathcal{X}}(\mathcal{G})$ the *empirical Rademacher complexity* of $\mathcal{G}$ conditioned on $\mathcal{X}$ [18]:

$$\widehat{\Re}_{\mathcal{X}}(\mathcal{G}) = \mathbb{E}_\sigma \left[ \sup_{g \in \mathcal{G}} \frac{1}{n} \sum_{x_i \in \mathcal{X}} \sigma_i g(x_i) \right].$$

It is known that $\widehat{\Re}_{\mathcal{X}}(\mathcal{G}) \leq C_w C_\phi / \sqrt{n}$ and thus $\Re_{n,q}(\mathcal{G}) = \mathbb{E}_{\mathcal{X}}[\widehat{\Re}_{\mathcal{X}}(\mathcal{G})] \leq C_w C_\phi / \sqrt{n}$ [18]. Then, letting $C_\mathcal{G} = C_w C_\phi$ completes the proof. $\qquad\square$