[Reviews · NeurIPS 2016]

Reviewer 1

Summary

The paper presents a theoretical analysis of when learning from positive and unlabelled data may be superior to learning from fully labelled data. The analysis relies on upper bounds of the estimation errors for various risks. Experiments largely agree with the theoretical analysis.

Qualitative Assessment

The basic problem studied in the paper concerns learning from data which is only partially labelled, but nonetheless doing better than with fully labelled data. In the particular scenario where one has a pool of unlabelled data in lieu of one of the classes, the paper seeks to quantify the impact this has on the estimation error of the learned classifier. Determining the degradation or lack thereof when learning from unlabelled data is interesting, and thus the paper seems well motivated. The machinery used to illustrate its messages are fairly standard -- the key quantities, namely the estimation errors for each scenario, are derived from a simple Rademacher analysis -- however, the final results appear novel, with implications worked through in various scenarios. The papers hinges on the simple facts that (a) different risks (for PN/PU/NU learning) may be seen as employing different weightings on individual risks for the positive and negative class, and (b) these ratings are reflected in appropriate terms for the estimation error. This, combined with the fact that in practice one is on the regime where unlabelled data is plentiful, is used to argue for the potential benefit is of learning from positive and unlabelled data. Quantifying the precise benefit that having a large amount of data from one "meta-class" buys you, and how it offsets the fact that this "meta-class" doesn't correspond to the actual underlying label, is nice. The estimation error analysis is fairly standard, and simply exploits the different weightings on the per-class risks. As a lacuna, the comparison of the various scenarios ends up being based on the upper bounds provided by the Rademacher analysis; this isn't ideal, as it implicitly assumes tightness of the bounds. This aside, the comparison of the of various settings in Thm 6 seems fairly unsurprising: one finds that depending on the relation of the number of observed samples to the base rate, either the positive and unlabelled or the fully labelled scenarios will be preferable. I admittedly did not get a lot out of the specific comparisons provided under various constraints on the number of samples; perhaps it would be better to introduce some practical scenario, and then try to motivate some particular relationship between the number of samples. Additionally, it seems natural to explicitly consider the case where n_- = n_u, where the bound for PN will dominate, to make clear that the benefit of PU data is purely from the ability to collect large numbers of samples from the unlabelled set. The asymptotic analysis in 3.3 aims to illustrate that with large amounts of unlabelled data, positive and on label learning can dominate the fully laden counterparts. The observation that one of PU or NU will dominate PN is intuitive in hindsight, but perhaps not obvious a-priori. I felt here that one could perhaps start off by considering the case where n_u = infty, before introducing the limit-based analysis. This would perhaps more directly hint at the power of the positive and unlabelled scenario. One limitation of the present analysis is that it restricts attention to losses satisfying the symmetry condition (2). It wasn't too clear why this was strictly necessary. What about losses satisfying ell(-1, v) - ell(1, v) = v, e.g. those considered in [5]? As I understand, the latter showed a risk equivalence for such losses as well. Is the point that in this case one doesn't have the exact same loss on the PU data, which perhaps complicates analysis somewhat? Other comments: - a citation for the empirical observation that learning from positive and unlabelled data performs better than positive and negative data seems prudent. - for the two sample positive and unlabelled setting, consider also citing Ward et al. Presence-only data and the EM algorithm. Biometrics, 2009. - the statement that the two sample scenario is more general than the one sample scenario is a bit unclear. The distributions that one observes in each scenario seem quite different. How exactly does one cast the noisy label scenario in terms of drawing from the positive conditional and the marginal? - there are a few improper uses of \mapsto instead of \to. - it wasn't entirely clear why one has the dependence on \pi in \hat{R}_pn. In practice surely one just uses 1/(n+ + n-) as a common scaling factor for the positive and negative risks, with no need to actually know the true value of the base rate. It seems that the paper employs this formulation in order to get an easy comparison to the bounds for the positive and unlabelled case? I note that otherwise Thm 3 for example would be nothing more than a standard Rademacher bound for the empirical risk, e.g. (Boucheron et al.), Thm 4.1. - it is said that [1] establishes that the 01 loss can be directly estimated in the positive and unlabelled setting; but that paper was only for the OS setting? And what is the specific theorem reference? - it could be made clearer in the theorem statements that one crucially relies on the loss satisfying the condition (2). - in the proof of corollary 5, it is unclear why the discussion proceeds onto the case where the function class is of the form (6). Isn't the proof complete at this stage? Is the idea is simply to show a direct proof for a special case? - Thm 6 could perhaps be demoted to Lemma 6? - the proof of lemma 8 could surely borrow from existing Rademacher analysis e.g. (Boucheron et al.), rather than invoking McDiamird and Talagrand from scratch.

Confidence in this Review

2-Confident (read it all; understood it all reasonably well)


Reviewer 2

Summary

In PU learning, we can only see positive and unlabeled examples (in NU learning, respectively, one sees only negative and unlabeled examples). This is different than more traditional learning, when one sees positive and negative examples. The paper focus is providing some theoretical justification for the effectiveness of PU learning. A natural, preciously proposed model is considered for how examples are drawn are considered. In this RS model, a set P of data is drawn from the density conditioned on positive examples and a set U is drawn from the overall examples. We can now apply learning algorithms to separate P and U the best we could, and this should provide a good separation also of positive and negative examples. When the loss function on negative and positive examples satisfies some symmetry condition (the authors use scaled ramp loss) then one can derive bounds in the absense of negative examples. The paper derives risk bounds under this model and certain assumptions (Lipschitz) on the quality of risk estimators that use different combinations of examples (PU, PN, NU). They show that with many U examples, the quality of a PU learner can be better than a PN learner. The paper includes some experiments. Artificial data with two Gaussians to generate the positive and negative examples and multiple "real-world" benchmark data sets. Not surprisingly, the worst-case theoretical bounds do not always agree with actual behavior. In particular, NU learning seems to be much worse than the supposedly symmetric case of PU learning. But the experiments do demonstrate the value of PU learning. Strong points: -- Theoretical analysis that explains to an extent the value of using many unlabeled exaples in some statistical settings (semi supervised learning) -- Good coherent presentation -- comprehensive experiments Weak points: -- smallish data sets in experiments

Qualitative Assessment

Strong points: -- Theoretical analysis that explains to an extent the value of using many unlabeled exaples in some statistical settings (semi supervised learning) -- Good coherent presentation -- comprehensive experiments Weak points: -- smallish data sets in experiments

Confidence in this Review

2-Confident (read it all; understood it all reasonably well)


Reviewer 3

Summary

In many learning situations, examples of one class are exclusively available and unlabeled data are abundant. This paper proposes Rademacher complexity bounds for the generalization error of classifiers trained with Positive and Unlabeled examples. By comparing the bound with the one obtained for the classical case of learning with positive and negative examples a compromise is derived of when the former should be preferred to the latter. The paper is theoretically sound, but with a complete set of simulations with convincing results. The mathematics involved rely on ingenious rewritting of the involved. The result is a very interesting contribution to the problem of PU.

Qualitative Assessment

I just have one remark that is the missing reference to the seminal work : @inproceedings{LetouzeyDG00, author = {Fabien Letouzey and Fran{\c{c}}ois Denis and R{\'{e}}mi Gilleron}, title = {Learning from Positive and Unlabeled Examples}, booktitle = {Algorithmic Learning Theory, 11th International Conference ({ALT})}, pages = {71--85}, year = {2000} } that studied the problem with regard to another tools than Rademacher complexity bounds.

Confidence in this Review

2-Confident (read it all; understood it all reasonably well)


Reviewer 4

Summary

The paper proposes to study the theoretical properties of PU learning and traditional binary classification. The paper tries to study the estimation error for the above two learning setting to get some results.

Qualitative Assessment

The paper contains some advantages and disadvantages. Advantages: 1. The sub-title of the paper is generally well organized. Disadvantage: 1. The paper is very difficult to understand and it is quite difficult to follow. One reason is that the notations of the paper are too many, and need re-organized to make the paper easier to follow. Another reason is that the paper introduces many abbreviations, e.g., P, N, TS, OS. That will hurt the readability of the paper. 2. I am not sure why the studied problem is of importance. The paper compares two learning scenarios, i.e., the PU learning and PN learning. This, however, is not practical in realistic cases. In realistic cases, if we obtain PN data, we could not be able to use PU learning algorithm. On the other side, if we obtain PU data, it is also impossible to use PN learning algorithm. So, the current study has a gap to real situations and the paper needs to figure out a more realistic setting. 3. The motivation of the studied problem is a bit week. The paper claims that PU could sometimes better than PN. However, from the paper, it is not clear how many evidences have been shown to prove it and what serious outcome will happen once it exists. Without those evidences, it is impossible to believe that such a phenomenon is of importance.

Confidence in this Review

2-Confident (read it all; understood it all reasonably well)


Reviewer 5

Summary

This paper investigates an interesting problem: comparing the risk of classification based on Positive-Negative (PN) learning and Positive-Unlabeled (PU) learning. Considering that in many real cases negative data are not easy to collect, the problem investigated in this paper is indeed important. In particular, this paper establishes the risk estimation of PU as well as NU learning schemes. This paper presents some unique contribution: such as proof on the classification-calibrated properties of the scaled ramp loss (Theorem 1).

Qualitative Assessment

My major concern about this paper is whether its claim is meaningful. The main conclusion presented by this paper is that PU and NU learning are likely to outperform PN learning when the number of U samples is sufficiently large (or even approximates infinite). This (somewhat surprising) result relies on Theorem 2 to 4, i.e. the risk estimation errors derived from Rademacher complexity. However, it is known that such risk estimation error bound is usually loose. So, one cannot safely and confidently conclude that PU learning is better or worse than PN learning based on such upper bounds. Can the authors explain the tightness of Theorem 2 to 4? Also can the authors specify the value of Rademacher complexity involved in the upper bounds? Another concern is about the experiments. The datasets used in the experiments are small-scale and dimensions of most of the datasets are small. In addition, the experiments on artificial data, the numbers of positive and negative samples are far from being balanced (45 vs. 5). Any explanation on such a choice?

Confidence in this Review

2-Confident (read it all; understood it all reasonably well)


Reviewer 6

Summary

The paper compares the quality of standard PN (positive vs. negative) learning with PU (positive vs. unlabeled). The measure of accuracy is "estimation errors". The paper cites previous work that showed that PU can outperform PN at times. The goal of the current paper is to explain this counterintuitive observation and derive when it holds. The measure of quality are "estimation errors" It appears that the results of this paper can contribute considerably to a theoretic understanding of the comparison between PN and PU learning. That is great and the question asked very interesting. A big concern of this paper is a lack of intuition for this result. The result is clearly counterintuitive, thus there should be simple way to explain why the result holds. Or better: when does it hold and does not hold, and what are the assumptions made by the paper (I could not follow the technical details) that allow this result to hold. The introduction (section 1) does not do an appropriate job in that. Also, how do the bounds on estimation errors translate, in practice, into labeling quality? What is the key assumption of the bounds that are used and the limit case of infinite labels?

Qualitative Assessment

I suggest the authors include high-level discussion of the counterintuitive results of the paper. What assumptions are used to come to the conclusion? How do bounds on estimation errors translate into actual labeling accuracy in practice?

Confidence in this Review

1-Less confident (might not have understood significant parts)